# Information about task duration influences energetic cost during split-belt adaptation and retention of walking patterns post-adaptation

Samantha Jeffcoat[1¤a], Adrian Aragon[1¤a], Andrian Kuch[1¤a], Shawn Farrokhi[1¤a], Andrew Hooyman[1¤a], Russell Johnson[2¤b], Natalia Sanchez[1,3¤a]*

1 Department of Physical Therapy, Crean College of Health and Behavioral Sciences, Chapman University, Irvine, California, United States of America, 2 Department of Physical Medicine and Rehabilitation, Northwestern University, Chicago, Illinois, United States of America, 3 Department of Electrical Engineering and Computer Science, Fowler School of Engineering, Chapman University, Orange, California, United States of America

¤a Current address: Department of Physical Therapy, Crean College of Health and Behavioral Sciences, Chapman University, Irvine, California, United States of America
¤b Current address: Department of Physical Medicine and Rehabilitation, Northwestern University, Chicago, Illinois, United States of America
* sanchezaldana@chapman.edu

## Abstract

Studies of locomotor adaptation have shown that adaptation can occur in short bouts and can continue for long bouts or across days. Information about task duration might influence the adaptation of gait features, given that task duration influences the time available to explore and adapt the aspects of gait that reduce energy cost. We hypothesized that information about task duration and frequency of updates influences adaptation to split-belt walking based on two competing mechanisms: individuals anticipating a prolonged adaptation period may either (1) extend exploration of energetically suboptimal gait patterns, or (2) adapt toward a more energy-efficient pattern earlier to maintain an energetic reserve. We tested three groups: N = 19 participants received minute-by-minute updates during a 10-minute adaptation duration (True group), N = 19 participants received no updates during a 10-minute adaptation duration and were misled to expect a prolonged 30-minute adaptation duration (False group), and N = 14 participants received one update halfway through a 10-minute adaptation duration (Control group). We measured step length asymmetry, leg work, and metabolic cost. Our results partially supported our hypothesis but did not confirm the underlying mechanisms. While step length asymmetry did not differ significantly between groups during adaptation, the True group generated a more effortful gait pattern with a greater increase in metabolic cost and higher work with the slow leg. Additionally, the True group showed no association between the different adapted gait variables such as step length asymmetry and metabolic cost, contrary to the Control and False groups. Finally, we observed that the False group showed greater

**Data availability statement:** A detailed R-markdown with statistical models and the data used for statistical analyses for this manuscript are available at: DOI 10.17605/OSF.IO/AW6XK https://osf.io/aw6xk/.

**Funding:** This work was funded by the National Center for Medical Rehabilitation Research R03HD107630, and the National Center for Advancing Translational Sciences R03TR004248 to N. Sanchez. The funders had no role in study design, data collection and analysis, decision to publish, or preparation of the manuscript.

**Competing interests:** The authors have declared that no competing interests exist.

retention of the split-belt aftereffects than the Control and False groups. Thus, adapted locomotor and energetic patterns are influenced by information about task duration, indicating that Information about task duration should be controlled for, or can be manipulated to elicit different efforts during adaptation.

## Introduction

Humans continuously adapt locomotor patterns in response to environmental conditions, such as changes in walking terrain, footwear, and load carriage [1–3]. In laboratory settings, split-belt treadmills have been used to study locomotor adaptation, where individuals walk on a treadmill with two separate belts (one for the right limb and one for the left limb) that can be independently controlled to move at different speeds [4,5]. Split-belt walking leads to immediate changes in intralimb coordination and gradual changes in interlimb coordination [5], which are attributed to distinct neural control mechanisms [5,6]. A vast amount of literature has shown that the neuromotor system can adapt to the split-belt's asymmetric environment and retain some of the adaptive changes in locomotion in the short and long term [5,7–10]. The kinematic, kinetic, and physiological changes that occur during short locomotor adaptation bouts of 10–15 minutes [11–21], during long adaptation bouts of 45 minutes [22], or during split-belt adaptation across multiple days [23] have been characterized; it is well established that individuals adapt from walking with asymmetric step lengths toward more symmetric step lengths, reduce positive work by the legs, and reduce metabolic cost during adaptation. However, it is not yet known whether these adapted gait patterns are influenced by the information about different adaptation durations.

Empirical [24–26], and simulation studies [27,28] have shown that different cost functions are optimized at different points in the adaptation process: the early stages of locomotor adaptation are associated with the regulation of dynamic balance control, whereas the later stages of locomotor adaptation are associated with reductions in energetic cost. A recent study reported that during a 20-minute split-belt walking protocol, measures of balance adapted in less than a minute, whereas metabolic cost and mechanical work adapted in ~4–6 minutes [25]. Our previous work showed that even after 45 minutes of continuous split-belt walking, individuals continued to adapt their gait pattern and achieve additional reductions in mechanical work and metabolic energy cost [22]. We also estimated from this prolonged adaptation bout that it takes around 1600 strides or close to 30 minutes to adapt step length asymmetry completely [22]. Adaptation has also been studied using exoskeletons [29–33]. A previous study assessing adaptation to an ankle exoskeleton showed that muscle activation patterns adapted faster than step frequency, and that the variability in step frequencies and muscle activation patterns decreased during adaptation [31]. This higher variability when first exposed to a novel locomotor adaptation task is thought to correspond to exploration of different aspects of the walking pattern to identify strategies that support energetic cost reduction [31,32]. Similarly, other studies using exoskeletons showed different timescales of adaptation to different exoskeleton parameters

[29]. Thus, given that multiple domains of locomotor adaptation change over different timescales, we propose that task duration will impact the way people adapt to a novel locomotor task.

Since energetic cost is optimized during the later stages of adaptation, information about a longer task duration might influence the adaptation of the aspects of gait underlying this energetic cost reduction. As individuals adapt their walking pattern to the split-belt treadmill, they reduce step length asymmetry, positive mechanical power, and positive mechanical work generated by their legs [11,19,20,22,24,25], which is associated with a reduction in metabolic cost [11,22]. In fact, studies have shown that walking patterns can change continuously, over different timescales, even for small energetic savings [32–34]. Thus, it may be the case that planning to sustain a task for a shorter or longer time may influence how the walking pattern is adapted to reduce energy cost.

In addition to task duration itself, the amount and frequency of information provided about that duration may also influence motor adaptation and control. For example, studies on feedback schedules have shown that infrequent feedback can impair immediate performance but enhance long-term learning [35]. Although our study does not provide performance feedback, it is possible that infrequent updates about task duration could produce similar effects. Moreover, motor control studies have demonstrated that when participants are given information about time constraints, they often select more effortful, energetically sub-optimal walking speeds to ensure task completion [36]. While in our study the time and speeds are fixed, information about time duration may still influence effort. This idea aligns with evidence that the central nervous system modulates muscle recruitment and power output based on expected exercise duration [37,38], and with the "end spurt" phenomenon, where effort increases as individuals approach the end of a task [39,40]. Given that the influence of task duration information on split-belt adaptation has not been directly studied, our work offers a novel contribution to the motor adaptation literature by exploring how time information and frequency of time updates may shape locomotor adaptation.

The primary aim l of our study was to determine if information about task duration affects the adaptation of locomotor patterns. We tested the hypothesis that individuals provided with minute-by-minute information of a 10-minute adaptation duration (True group: True task duration information with frequent updates on time remaining) will have a different adapted locomotor pattern with different metabolic cost than those individuals who were informed that they will sustain a locomotor adaptation task for a prolonged time of 30 minutes but who will actually sustain it for 10 minutes (False group: False task duration information with no updates of time remaining). We compared our two experimental groups to a Control group, who received true information about task duration before the start of the adaptation trial, without the minute-by-minute updates and with only a time update at the halfway point, consistent with previous work [22,24–26,34]. Two contrasting mechanisms can explain this metabolic and locomotor pattern differential: Mechanism 1) given that the split-belt task is only of moderate intensity, the False group might spend more time exploring suboptimal locomotor patterns as they prepare to sustain the task for longer, whereas the True group will aim to reach a less costly pattern within the allotted time; if mechanism 1 drives adaptation, we hypothesize that the False group will show less adaptation of step length asymmetry, higher metabolic cost, and more work by the legs compared to the True and Control groups. Under mechanism 1, we also hypothesize that the True group would be less costly than the Control group. Mechanism 2) The False group might adapt toward a generally less costly pattern than the True group to be able to maintain an energetic reserve needed to sustain the task longer [37,38,41]; if mechanism 2 drives adaptation, we hypothesize that the False group will show greater adaptation, lower metabolic cost, and less work by the legs compared to the True and Control groups. Under mechanism 2, we expect no differences between the True and Control groups. Since our previous work showed that adaptation duration influences the duration but not the magnitude of the aftereffects during post-adaptation [22], and all groups will adapt for the same duration, we hypothesize that the locomotor pattern during post-adaptation will not differ between groups. A better understanding of how information about task duration and updates provided during adaptation tasks influence adaptive processes is vital in the design of training schedules aimed at retraining walking behaviors as part of rehabilitation interventions [42,43] or via the use of assistive devices [29,33,44,45].

## Materials and methods

Since our hypothesis is based on differences in metabolic cost, we assessed the effect size of detecting a difference in metabolic cost after 10 minutes of adaptation. We obtained metabolic cost measures in N = 15 individuals from a previous study that assessed split-belt adaptation duration of 45 minutes [22]. We extracted data during split-belt walking between minutes 1–2 (early adaptation) and 9–10 (late adaptation in the experimental protocol used here). We obtained the mean and standard deviation of the metabolic cost at these points to calculate the effect size needed to detect a reduction in metabolic cost for 10 minutes of adaptation. These data provided an effect size of 0.76. With a sample of 16 individuals in the True and False groups, we would have a power of 0.82 to detect a reduction in metabolic cost.

To assess differences between the True group and the False group relative to the locomotor pattern described in prior adaptation studies, we also collected a Control group post-hoc, which, like previous work, was informed about a 10-minute task duration at the beginning of the split-belt adaptation trial and only received one update halfway during adaptation. We collected a Control group of N = 14 participants, with a sample size similar to previous studies that showed a reduction in energetic cost or mechanical work during adaptation (N = 14 in [46], N = 11 in [11] and N = 14 in [19]). This group also allowed us to ensure that any potential differences in our True and False groups are due to our experimental manipulation of providing different information on task duration and frequency of updates in the False or True groups, and not due to potential confounding effects of our experimental environment or analyses.

Participants between the ages of 18–35 were recruited for this study. Participants were excluded if they had a history of lower extremity surgery or if they had any lower extremity injury such as a fracture or sprain in the last two years, performed more than 10 hours of weekly exercise, reported current cough, cold, congestion, or any respiratory ailment that affected breathing, had previous experience walking on a split-belt treadmill, or had a history of neurological disorders or severe head trauma. The Chapman University Institutional Review Board approved all experimental procedures, with approval number IRB-23–189, and participants provided written informed consent before testing. All study aspects conformed to the principles described in the Declaration of Helsinki.

Data were collected at the Gait Behavior Lab at Chapman University's Department of Physical Therapy using a Gait Real-time Analysis Interactive Lab (GRAIL) system (Motek Medical Base.V., Houten, Netherlands). The GRAIL is equipped with 10 motion capture cameras (Vicon Motion Systems Ltd, Oxford, UK), and an instrumented dual-belt treadmill (Motek Medical Base.V). We used a metabolic cart to measure energetic expenditure (TrueOne 2400, Parvomedics, Salt Lake City, UT, USA). Participants were instructed not to eat or have caffeine at least two hours before the experiment. Before each experiment, we calibrated the motion capture system and the metabolic cart per the manufacturer's specifications. During all walking trials, participants wore a harness to prevent falls, without providing body weight support. Participants did not hold on to a handrail at any point in the experiment.

### Experimental Protocol

After informed consent, participants were fitted with a mask covering their nose and mouth to measure their breathing (Fig 1A). First, participants' resting caloric expenditure was measured while standing for six minutes. Then, participants underwent three walking trials with breaks between each trial until metabolic cost reached standing baseline levels to ensure resting conditions before continuing. The first trial, Baseline (Base), had participants walking for six minutes with the belts under each leg moving at 1.0 m/s. The second trial (Split) had participants walking for 10 minutes with the left belt moving at 1.5 m/s (fast) and the right belt at 0.5 m/s (slow) (Fig 1B, C). We chose 10 minutes of split-belt adaptation as this duration allows us to observe adaptation of spatiotemporal gait measures [5], metabolic cost [11,46], and work by the legs [19]. Participants were assigned to either the True or False groups via a coin flip to ensure randomization. The True group participants performed the split-belt adaptation trial with true information of how long the trial would last, and they were given an update on the time remaining every minute. The False group participants were given the misleading

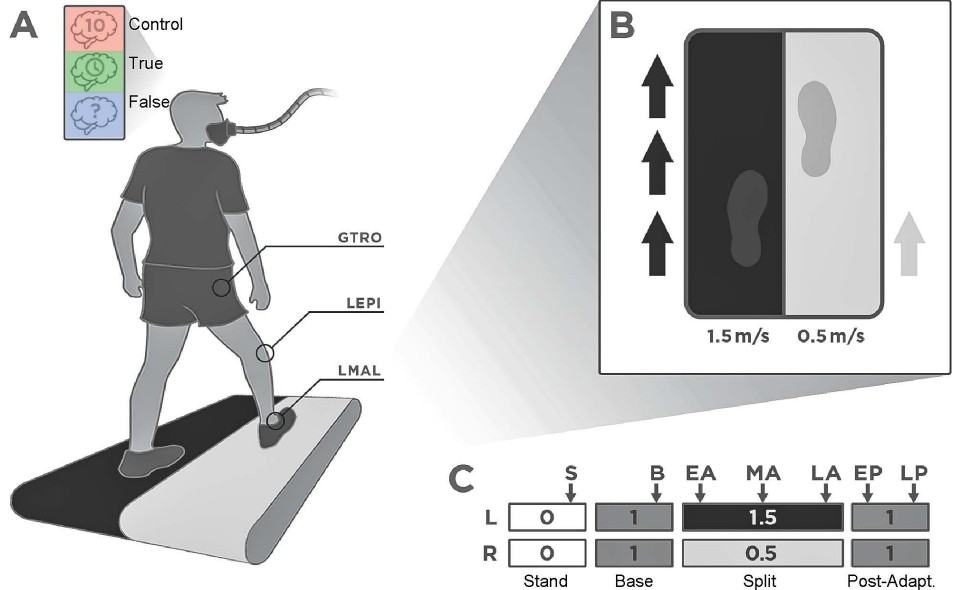

**Fig 1. Experimental setup and protocol.** A) Experimental setup showing experimental groups, metabolic mask, and marker location to calculate spatiotemporal variables. From top to bottom, the groups are the Control group (red), the True detailed time information group (green), and the False time information group (blue). B) Split-belt configuration. For all participants, the left belt was set at 1.5 m/s and the right belt at 0.5 m/s. 52 participants reported the right leg as the leg they would use to kick a ball. C) Experimental protocol. Bars indicate each belt, and numbers indicate belt speed. Bar width indicates the duration of each experimental condition. Participants reported their rate of perceived exertion during the time points identified by the arrows. S: Standing. B: Baseline. EA: Early adaptation. MA: Mid-adaptation. LA: Late adaptation. EP: Early post-adaptation. LP: Late post-adaptation. GTRO: Greater Trochanter. LEPI: Lateral Epicondyle. LMAL: Lateral Malleolus.

instruction verbatim: "You will be walking until we obtain the data needed, which can take about 30 minutes". Participants in the False group were not given updates on time elapsed or time remaining, and they also walked for 10 minutes. At the end of the 10th minute, we stopped the treadmill and told participants that we had collected all the necessary data. Participants in the Control group were informed that the task would last 10 minutes and were only given an update halfway that they had five minutes remaining. The final trial (Post-adaptation) had participants walking again during a post-adaptation trial for 6 minutes with the belts under each leg moving at 1.0 m/s. All participants received the same instructions for the post-adaptation trial, indicating that this was the final trial and that it would last six minutes.

Participants were asked to rate their level of exertion via the modified Borg Rating of Perceived Exertion (RPE) [47] using their fingers, since they were wearing the metabolic mask, to indicate on a scale of 1–10 which level of the RPE table they perceived. During the Base trial, participants reported RPE at the five minute mark. During the Split trial, they were asked to rate their level of exertion at the one (early adaptation – EA), five (mid-adaptation – MA), and nine minute (late adaptation – LA) marks. During the tied post-adaptation trial, participants rated their RPE at the one (early post-adaptation – EP) and five minute marks (late post-adaptation – LP) (Fig 1C).

## Data acquisition

We recorded marker data to calculate spatiotemporal variables and force data to calculate work by the legs. The positions of reflective markers located bilaterally on the greater trochanter, lateral epicondyle of the femur, and lateral malleolus were recorded at 100 Hz (Fig 1A). Force plates embedded into the split-belt treadmill recorded the ground reaction forces generated by each leg with a sampling frequency of 1000 Hz. Metabolic cost was recorded by determining the rates of

oxygen consumption (VO$_2$) and carbon dioxide production (VCO$_2$). We also collected heart rate (HR) as a measure of cardiovascular intensity using a Polar H10 chest strap monitor (Polar Electro Inc., Bethpage, NY, USA), synchronized to the metabolic cart. Metabolic cost and kinematic data were collected for all participants. Heart rate data were missing in six participants due to Bluetooth connectivity issues (three in the Control group, one in the True group, and two in the False group). Force data were collected from all participants except one in the False group, for whom force data were corrupted, which hindered performing mechanical calculations. Data for all participants, as available, were included in all analyses.

## Data processing and analysis

We used custom-written code in MATLAB R2023b (Mathworks, Natick, MA, USA) for all data processing and analyses as in previous work [22,34]. We estimated the energy consumed using the standard Brockway equation to obtain metabolic rate [48]. We subtracted each participant's standing metabolic rate from each walking trial. Thus, all metabolic rate values presented here are net metabolic rates. A fourth-order low-pass digital Butterworth filter smoothed marker data and ground reaction force data using cut-off frequencies of 10 Hz and 20 Hz, respectively. We calculated step lengths using marker data, defined as the distance between the lateral malleoli markers at the instance of peak anterior position of the respective marker. Step length asymmetry was then defined as follows:

$$SLA = \frac{SL_{fast} - SL_{slow}}{SL_{fast} + SL_{slow}}$$

(1)

with $SL_{fast}$ representing the step length at heel strike of the fast leg, and $SL_{slow}$ representing the step length at heel strike of the slow leg. We calculated leading and trailing limb placement for each limb as the fore-aft distance of the ankle markers relative to the midpoint between markers placed bilaterally on the greater trochanters.

We estimated the mechanical work performed by the legs using an extension of the individual limbs method [19,20,22,34,49,50]. This method approximates the legs as massless pistons and the entire body as a point mass. We measured individual leg ground reaction forces from the instrumented treadmill, and segmented forces into strides using a vertical ground reaction force threshold of 32 N [19] to identify the beginning and end of each stride. We calculated the mediolateral, fore-aft, and vertical center of mass velocities by integrating the center of mass accelerations. We then calculated mechanical power as the dot product between the center of mass velocity and the ground reaction force from each leg. To determine the total positive and negative work performed by a leg or a belt, we calculated the time integral of the positive or negative portion of the mechanical power over the stride cycle. Work was normalized by leg length and mass to derive unitless mechanical work. We calculated leg length as the distance between the greater trochanter markers and the ground during standing.

We derived variability during adaptation as in a previous study [31]. We determined the variability by applying a high-pass filter with a cut-off frequency of 0.033 steps$^{-1}$, and then calculated the standard deviation of this filtered signal during the initial, middle, and final minute of the trial. The variability was expressed as a normalized value relative to Base variability. We assessed variability in variables that represent different constructs of adaptation: step length asymmetry, which is used to track adaptation, net work due to its relationship to metabolic cost, and step width as a proxy for balance.

## Statistical analyses

Statistical analyses were run in R version 4.2.2 and MATLAB R2023b. We assessed data for normality using the Shapiro-Wilk test. We compared age, height, and weight between the three experimental groups using one-way ANOVA or Kruskal-Wallis tests for not normally distributed data.

We obtained average values for step length asymmetry, step lengths, step width, and positive and negative work for each leg for the last five strides of the Base trial. For EA, MA, and LA we averaged the first, middle, and last five strides of

the Split trial [26]. For the Post trial, we averaged the first five and last five strides (EP, LP, respectively). These are close to the time points when we obtained RPE and average measures of metabolic cost.

The metabolic power for the Base condition was computed by averaging the power for the last two minutes. The metabolic power for EA, MA, and LA was averaged between the second and third minutes, the fifth and sixth minutes, and the ninth and tenth minutes, respectively, of the Split condition to match the RPE reporting periods (Fig 1C). We used the second instead of the first minute for metabolic data as the average duration to reach peak metabolic power is around one minute [11,51]. Finally, the metabolic power for the Post-adaptation condition was averaged between the second and third minutes and the fifth and sixth minutes of this condition to obtain average values of metabolic cost for EP and LP, respectively.

We tested differences during the Base condition between groups using linear models with group (Control, True, False) as a categorical predictor and all outcome variables as a continuous response, to assess if there are baseline differences between groups that could have influenced our results. As the goal of our analysis was to observe differences within the Split and Post-adaptation conditions between groups, we ran separate models for these conditions. For the Split condition, we used linear models with time (EA, MA and LA) and group (Control, True, False) as categorical variables. We also included the value for the respective outcome measured during Base as a covariate, and its interaction with group and time, given that we observed that the response during the adaptation task depended on baseline walking characteristics. For the Post-adaptation condition, we used linear models with time (EP and LP) and group (Control, True, False) as categorical variables. We also included the value for the respective outcomes measured during Base as a covariate and its interaction with group and time. The models had the general form of:

$$x_{Split/Post} \sim \text{ Group} * \text{Time} + x_{Base} * \text{Time} + x_{Base} * \text{Group} \tag{2}$$

where $x$ is the scaled value measured during either the split-belt task or post-adaptation for any of the following outcomes: metabolic cost, step length asymmetry, step width, positive and negative work by each leg. Results for heart rate and rating of perceived exertion, step lengths, and step times are included in the Supporting Information. The reference level for the linear models for group was set as Control group, as we are comparing the True and False groups to a protocol that is comparable to that used in the split-belt literature. The reference level for the linear models for time was set as EA for the split-belt models and EP for the post-adaptation models. We assessed residual plots for each model to ensure that they were normally distributed. Post-hoc comparisons were corrected via the false discovery rate. We also assessed individual correlations between metabolic cost, work, step length asymmetry and step lengths between groups. To allow comparison of correlation coefficients for different sample sizes, we used a Fisher transformation of the resulting correlation coefficients.

To assess differences in variability, which we use here to quantify exploration, we used a linear model of the form:

$$variability_{x\ Split/Post} \sim \text{ Group} * \text{Time} \tag{3}$$

Where $x$ is either step length asymmetry, step width or net work. Since the variability data were normalized to Base, we did not use Base as a covariate. The reference level for this model was the Control group during EA.

## Results

N = 52 participants volunteered for this study: N = 14 for the Control group, N = 19 for the True group, and N = 19 for the False group. No significant differences were observed in participant demographics between groups (Table 1).

We did not observe significant differences in any of our outcome variables between groups during the Base condition that could have influenced the Split or Post-adaptation results (p > 0.352, Table 2).

**Table 1. Participant demographics for both experimental groups are reported as mean ± standard deviation.**

| Group | Control | True | False | |
|---|---|---|---|---|
| N | 14 | 19 | 19 | |
| Age (yrs) | 23.2 ± 3.4 | 24.8 ± 3.4 | 25.7 ± 4.7 | p = 0.191 |
| Height (cm) | 170 ± 10 | 167 ± 8.2 | 167 ± 9.2 | p = 0.446 |
| Weight (kg) | 70.2 ± 12.9 | 73.0 ± 13.1 | 67.6 ± 13.6 | p = 0.458 |
| Sex | 8F/6M | 13F/6M | 12F/7M | p = 0.801 |

M: Male, F: Female. P-values for one way ANOVA and for Chi-Square tests show no differences in group demographics.

**Table 2. Biomechanical and physiological characteristics for participants in the three groups during the Base trial walking with belts tied at 1.0 m/s.**

| Group | Control | True | False | |
|---|---|---|---|---|
| Metabolic Cost (W) | 2.26 ± 0.51 | 2.47 ± 0.42 | 2.43 ± 0.51 | p = 0.487 |
| Heart Rate (bpm) | 100 ± 14 | 100 ± 12 | 94 ± 12 | p = 0.352 |
| RPE | 2.0 ± 0.75 | 1.75 ± 1.0 | 2.0 ± 0.92 | p = 0.675 |
| Step Length Left (mm) | 514 ± 36 | 509 ± 28 | 511 ± 24 | p = 0.900 |
| Step Length Right (mm) | 509 ± 46 | 500 ± 24 | 508 ± 29 | p = 0.682 |

Data are reported as means ± standard deviation. P-values for linear models using group as a categorical predictor. bpm: beats per minute. RPE: rating of perceived exertion 1–10 scale.

## Information about task duration did not influence step length asymmetry during split-belt adaptation

We hypothesized different adaptation patterns between groups. The linear model assessing step length asymmetry as a function of group, time and baseline asymmetry during split-belt adaptation returned a significant intercept ($\beta_0 = -0.30$, 95%CI [−0,36, −0.25], p < 0.001), and a significant main effect of time ($\beta_{TimeMA} = 0.24$, 95%CI [0.16, 0.32], p < 0.001 and $\beta_{TimeLA} = 0.28$, 95%CI [0.20, 0.36], p < 0.001) indicating a reduction in asymmetry during adaptation across groups. We also observed an effect of the False group ($\beta_{GroupF} = -0.08$, 95%CI [−0.15, 0.01], p = 0.036), indicating that the False group was slightly more asymmetric but this more negative asymmetry was offset by a marginal interaction between the False group and MA ($\beta_{GroupF:TimeMA} = 0.09$, 95%CI [−0.01, 0.20], p = 0.072) and the False group and LA ($\beta_{GroupF:TimeLA} = 0.10$, 95%CI [−0.01, 0.20], p = 0.068) (adjusted $R^2 = 0.66$) (Fig 2A–B). Contrary to our hypothesis, we observed a similar adaptation of step length asymmetry across groups.

The linear model assessing step width as a function of group, time, and baseline asymmetry during split-belt adaptation returned a significant intercept ($\beta_0 = 281$, 95%CI [267, 294], p < 0.001) and a significant effect of baseline step width ($\beta_0 = 19$, 95%CI [10, 28], p < 0.001) but no effect of group or time (Fig 2C), indicating that individuals did not adapt step width (adjusted $R^2 = 0.34$). We did not see differences between groups for fast and slow step lengths, swing times, and limb placements (Supporting Information).

## Information about task duration influenced metabolic cost during split-belt adaptation

We hypothesized differences in metabolic cost between groups during adaptation. The linear model assessing metabolic cost as a function of group, time, and baseline cost during split-belt adaptation returned a significant intercept ($\beta_0 = 3.30$, 95%CI [3.04, 3.96], p < 0.001), a significant main effect of baseline metabolic cost ($\beta_{Base} = 0.30$, 95%CI [0.13, 30.48], p < 0.001), and a significant main effect of LA ($\beta_{TimeLA} = -0.39$, 95%CI [−0.76, 0.03], p = 0.035) indicating that across all groups, participants reduced cost from EA to LA. The model returned a significant interaction between the True group and

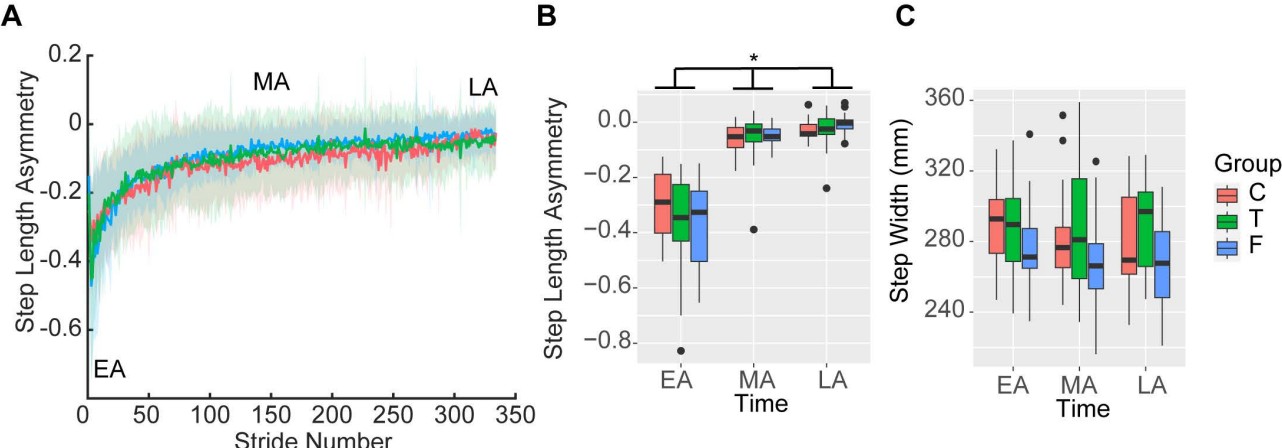

**Fig 2. Step length asymmetry and step width during split-belt adaptation.** A) Step length asymmetry timeseries during the split-belt walking between the control (Control – red), true information about task duration (True – green), and false information of task duration (False – blue) groups. Time windows indicating EA, MA, and LA are indicated in the figure. B) Distribution of average step length asymmetry between groups during the five strides corresponding to EA, MA, and LA. Across groups, asymmetry decreased from EA to MA and LA. (p<0.001) C) Distribution of average step widths. No differences over time or between groups were observed. Abbreviations as in Fig 1.

baseline metabolic cost ($\beta_{GroupT:BaseMet}$=0.32, 95%CI [0.11, 0.52], p=0.002) (adjusted $R^2$=0.66). Estimated marginal trends of the slope between baseline metabolic cost with split-belt metabolic cost by groups show a greater slope in the True group compared to both the Control group (slope True=1.27 vs slope Control=0.62, p=0.007) and the False group (slope False=0.84, p=0.044), indicating a greater increase in the True group's metabolic cost relative to baseline metabolic cost during split-belt adaptation (Fig 3A–C). These results support our hypothesis of different costs between groups, but does not support either hypothesized mechanism. These different costs were due to an increased rate of change from baseline metabolic cost in the True group, contrary to our two mechanistic hypotheses. Heart rate and RPE results are presented in the S1 File.

### Information about task duration influenced positive work by the slow leg during split-belt adaptation

We hypothesized that less mechanical work would be performed in the less metabolically costly group. The linear model assessing positive work by the fast leg as a function of group, time and baseline work during split-belt adaptation returned a significant intercept ($\beta_0$=0.036, 95%CI [0.032, 0.040], p<0.001), a significant main effect of baseline work by the fast leg ($\beta_{Base}$=0.005, 95%CI [0.002, 0.008], p<0.001), an effect of time ($\beta_{TimeMA}$=−0.011, 95%CI [−0.017, −0.006], p<0.001 and $\beta_{TimeLA}$=−0.013, 95%CI [−0.018, −0.008], p<0.001) but no effect of group (adjusted $R^2$=0.52). Contrary to our hypothesis, the model suggests that all individuals across groups similarly reduced positive work by the fast leg during adaptation (Fig 4A, C).

The linear model assessing positive work by the slow leg as a function of group, time and baseline work during split-belt adaptation returned a significant intercept ($\beta_0$=0.013, SE=0.0014, p<0.001), a significant main effect of baseline work by the slow leg ($\beta_{Base}$=0.003, SE=0.0010, p=0.010), and a significant main effect of the True group ($\beta_T$=0.005, SE=0.0018, p=0.012). These results indicate that positive work by the slow leg in the True group was significantly higher during adaptation than in the Control and False groups (adjusted $R^2$=0.34) (Fig 4B, D). Thus, supporting our hypothesis and mechanism 2 of a lower cost in the False group, we observed differences in positive work by the slow leg between groups, with the less costly groups (Control and False) generating less positive work. However, consistent with the metabolic cost findings, this was due to increased positive work with the slow leg in the True group. Contrary to our hypothesis,

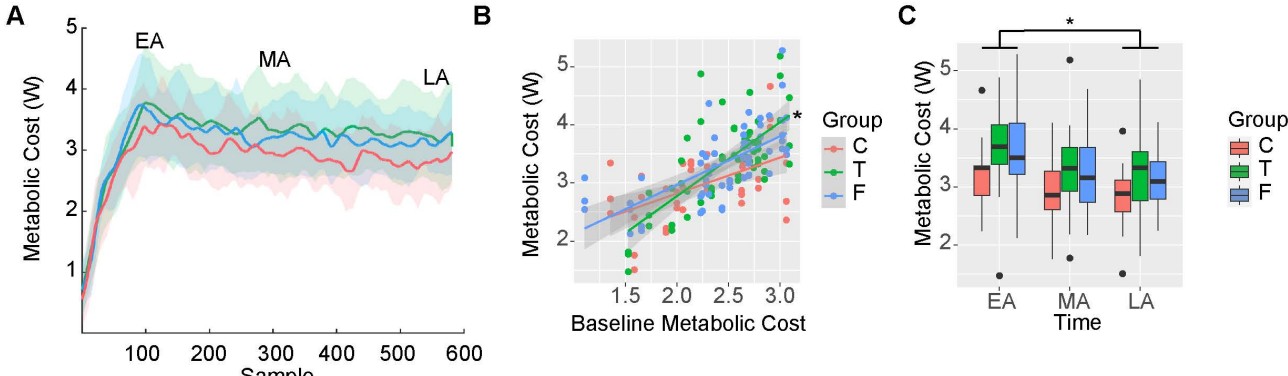

**Fig 3. Metabolic cost, heart rate, and perception of effort during split-belt adaptation.** A) Net metabolic power timeseries during the split-belt walking between the control (Control – red), true information of task duration (True – green) and false information of task duration (False – blue) groups. Time windows indicating EA, MA, and LA are indicated in the figure. B) Slopes of the relationship between metabolic cost during baseline and metabolic cost during EA, MA and LA. A significantly greater slope was observed in the True group compared to both the Control and False groups (adjusted p < 0.050). C) Average metabolic cost during EA, MA, and LA. A significant reduction in metabolic cost from EA to LA was observed across groups. Other abbreviations as in Fig 1 and Table 2.

we observed differences between the Control and True groups due to the increased work in the True group. Results for negative work are presented in the S1 File.

### Information about the duration of split-belt adaptation influenced the association between biomechanical and metabolic changes

We found a significant association between the change in step length asymmetry and the change in metabolic cost for the Control group (r = −0.55, Fisher transformed r = −0.61, p = 0.043) and for the False group (r = −0.51, Fisher transformed r = −0.56, p = 0.027). We did not observe an association between metabolic cost and asymmetry in the True group (r = −0.31, Fisher transformed r = −0.32, p = 0.203) (Fig 5A).

We found a significant association between the reduction in positive work by the fast leg and change in metabolic cost for the False group (r = 0.67, Fisher transformed r = 0.81, p = 0.002), but not in the Control (r = 0.50, Fisher transformed r = 0.54, p = 0.071) or True (r = 0.41, Fisher transformed r = 0.44, p = 0.079) groups (Fig 5B). We did not find an association between the reduction in positive work by the slow leg and change in metabolic cost (Fig 5C).

We found a significant association between the change in step lengths of the fast and slow leg and the change in positive work by the fast leg in the False group (r = −0.69, Fisher transformed r = −0.83, p = 0.002 for fast step lengths and r = 0.55, Fisher transformed r = 0.62, p = 0.017, for the slow leg) (Fig 5D–E). We observed a significant association between the change in step lengths of the fast leg and positive work by the fast leg in the True group (r = −0.77, Fisher transformed r = −1.02, p < 0.001) (Fig 5D). We did not observe a significant association between the change in step lengths and work in the Control group or change in slow step lengths and work in the True group.

### Information about task duration influenced the exploration of walking patterns during adaptation

The linear model assessing variability in step length asymmetry as a function of group and time returned a significant intercept ($\beta_0$ = 4.66, 95%CI [3.51, 5.81], p < 0.001), a significant effect of time ($\beta_{TimeMA}$ = −2.52, 95%CI [−4.15, −0.89], p = 0.003 and $\beta_{TimeLA}$ = −2.51, 95%CI [−4.14, −0.88], p = 0.003), a significant effect of group ($\beta_{GroupT}$ = 2.40, 95%CI [0.88, 3.91], p = 0.002 and $\beta_{GroupF}$ = 2.73, 95%CI [1.22, 4.25], p < 0.001) and a significant interaction between time and group ($\beta_{TimeLA:GroupT}$ = −2.42, 95%CI [−4.57, −0.27], p = 0.027, $\beta_{TimeMA:GroupT}$ = −3.14, 95%CI [−5.29, −0.99], p = 0.004,

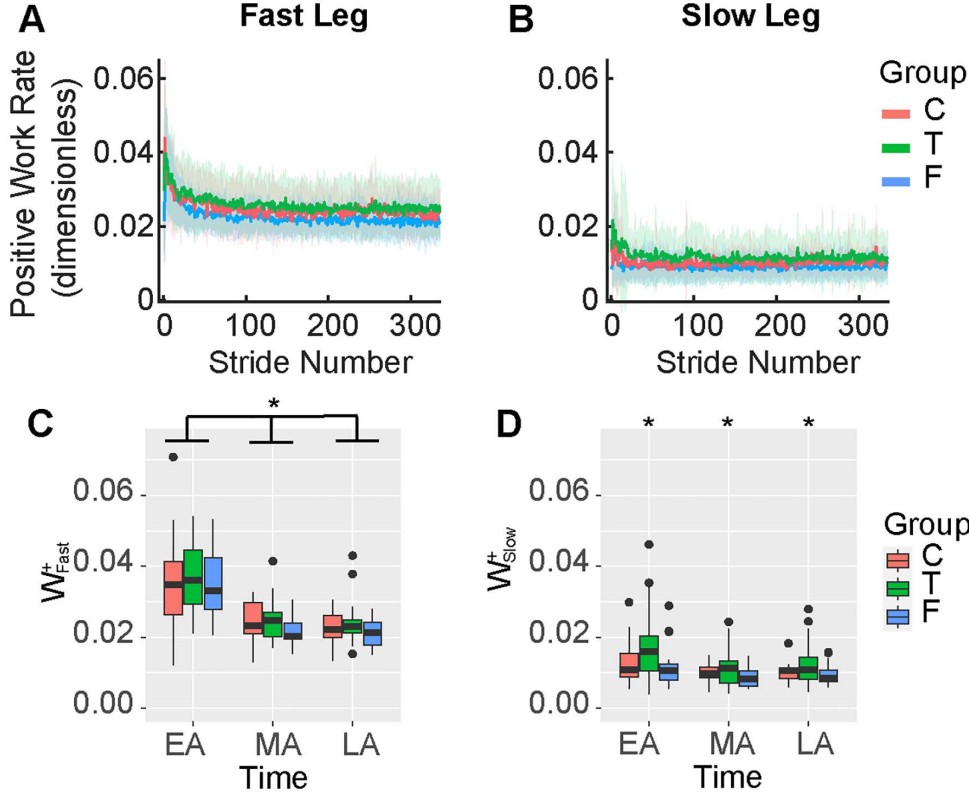

**Fig 4. Positive work by the fast and slow legs during split-belt adaptation.** A) Timeseries of positive work by the fast leg. B) Timeseries of positive work by the slow leg. C) Distribution of average positive work by the fast leg between groups during the five strides corresponding to EA, MA, and LA. We observed a significant reduction in work from EA to MA and LA across groups. D) Distribution of average positive work by the right leg between groups during the five strides corresponding to EA, MA, and LA. We observed significantly higher work in the True group (p = 0.012). $W^+$: positive work. Other abbreviations as in Figs 1 and 2.

$\beta_{TimeLA:GroupT}$ = −2.99, 95%CI [−5.13, −0.84], p = 0.007) (adjusted $R^2$ = 0.49). Post-hoc comparisons indicated significantly higher variability, indicating more exploration in the True and False groups during EA compared to the Control group (p < 0.005) (Fig 6A). We did not observe differences between MA and LA in variability as a measure of exploration. Thus, both the True and False groups had more exploration during EA than the Control group, contrary to one of our mechanistic hypotheses that the False group might spend more time exploring suboptimal locomotor patterns. The linear models assessing variability in step widths, and net work identified a significant main effect only of time but not of group, indicating that variability in step width and work decreased from EA to MA and LA similarly across groups (Fig 6B–C).

### Information about task duration during split-belt adaptation influenced retention of step length asymmetry during post-adaptation

We hypothesized no differences during post-adaptation between groups. The linear model assessing step length asymmetry as a function of group, time and baseline asymmetry during post-adaptation returned a significant intercept ($\beta_0$ = 0.24, 95%CI [0.20, 0.28], p < 0.001), a significant main effect of baseline asymmetry ($\beta_{Base}$ = 0.05, 95%CI [0.02, 0.08], p = 0.003), a significant main effect of LP ($\beta_{TimeLP}$ = −0.22, 95%CI [−0.28, −0.16], p < 0.001), a significant effect of False group ($\beta_{GroupF}$ = 0.12, 95%CI [0.07, 0.18], p < 0.001), a significant interaction between False group and LP ($\beta_{GroupF:TimeLP}$ = −0.11, 95%CI [−0.19, −0.03], p = 0.006) and a significant interaction between LP and baseline asymmetry

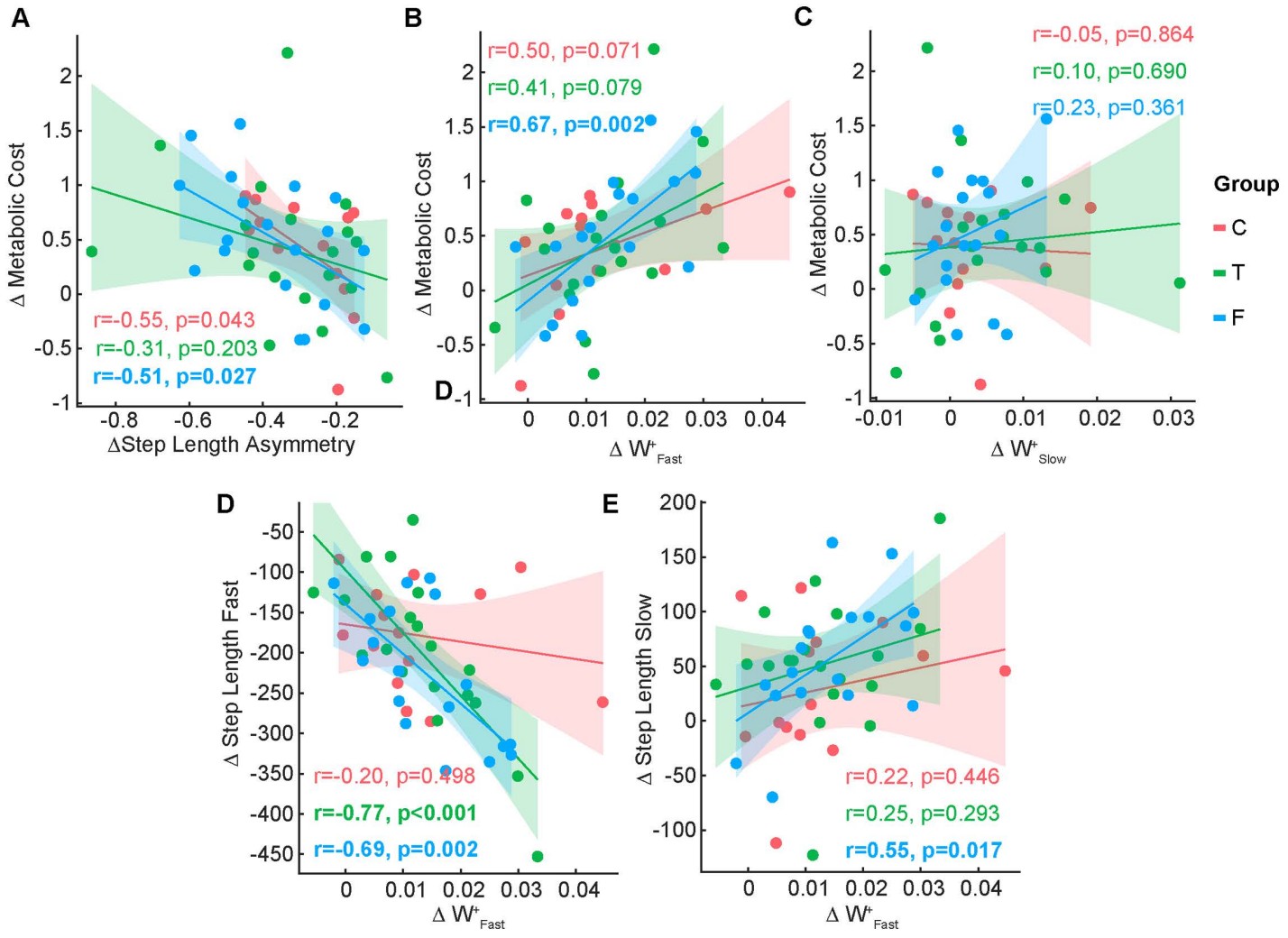

**Fig 5. Associations between biomechanical and metabolic measures.** A) Association between change in step length asymmetry and change in metabolic cost from EA to LA. B) Association between change in positive work by the fast leg and change in metabolic cost from EA to LA. C) Association between change in positive work by the slow leg and change in metabolic cost from EA to LA. D) Association between change in positive work by the fast leg and change in step lengths by the fast leg from EA to LA. E) Association between change in positive work by the slow leg and change in step lengths by the fast leg from EA to LA. Abbreviations as in Figs 1 and 2.

($\beta_{Base:TimeLP}$ = −0.04, 95%CI [−0.07, −0.005], p = 0.022) (adjusted $R^2$ = 0.77). Overall, these results indicate a reduction in asymmetry from EP to LP for all groups. Surprisingly, we found a higher asymmetry in the False group compared to the Control (p < 0.001) and True groups (p < 0.001) during EP, indicating a greater after-effect of the split-belt (Fig 7C–D). This contradicts our hypothesis that the post-adaptation locomotor pattern would be similar between groups. The after-effect in the False group can be explained by changes in the right leg, previously on the slow belt (Figs 4–5). The linear model assessing right step lengths as a function of group, time, and baseline step lengths during post-adaptation returned a significant intercept ($\beta_0$ = 347, 95%CI [318, 376], p < 0.001), a significant main effect of baseline step lengths ($\beta_{Base}$ = 38, 95%CI [20, 56], p < 0.001), a significant main effect of LP ($\beta_{TimeLP}$ = 157, 95%CI [117, 198], p < 0.001), a significant effect of the False group ($\beta_{GroupF}$ = −70, 95%CI [−107, −32], p < 0.001), and a significant interaction between the False group and LP ($\beta_{GroupF:TimeLP}$ = 63, 95%CI [10, 116], p = 0.043) (adjusted $R^2$ = 0.77). Overall, these results indicate that step length on

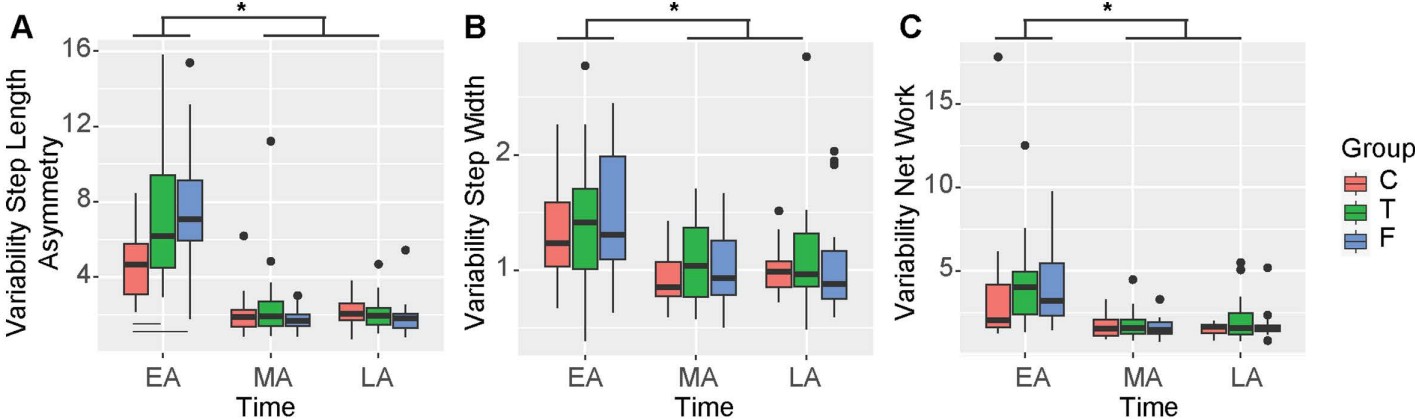

**Fig 6. Variability of gait features during adaptation.** A) We observed significant differences between groups in step length asymmetry variability during EA. Variability significantly decreased during MA and LA. No differences between groups were observed during MA and LA. B) Variability in step widths decreased over timepoints across all groups. C) Variability in net work decreased over timepoints across all groups. Abbreviations as in Figs 1 and 2.

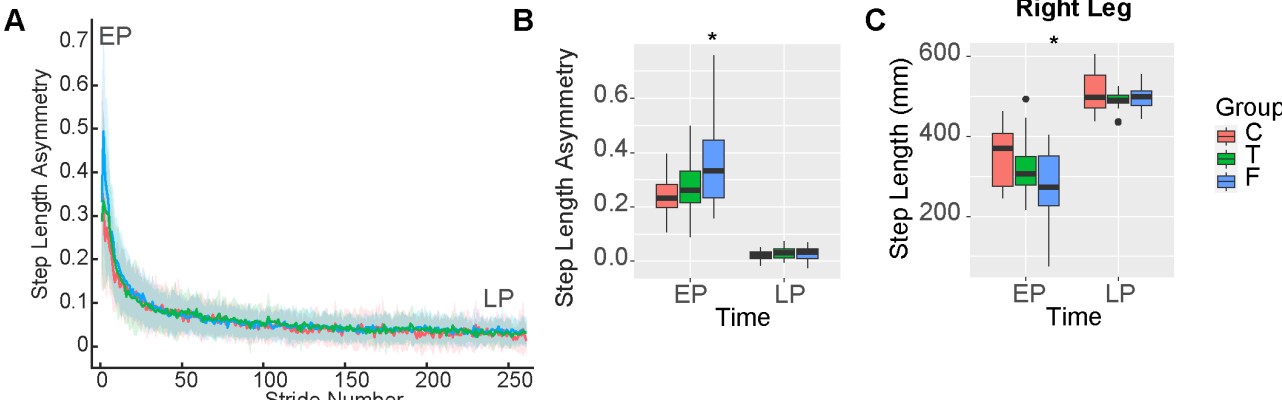

**Fig 7. Step length asymmetry and work during post-adaptation.** A) Step length asymmetry timeseries. B) Average step length asymmetry during EP, and LP. Participants in the False group had a greater asymmetry during EP compared to the Control and True groups (p<0.001). C) Average step lengths by the right leg during EP, and LP. Participants in the False group had a shorter step length with the right leg during EP compared to the Control and True groups (p<0.001). Abbreviations as in Figs 1 and 2.

the right leg remains shorter during EP in the False group compared to the Control group (p<0.001) and the True group (p<0.001) (Fig 7E). No differences between groups were observed for the left leg, which was previously on the fast belt. Results for metabolic cost and work are included in the S1 File.

## Discussion

Studies of motor adaptation have shown that adaptation begins within the first minute and continues even after 45 minutes of adaptation to split-belt walking [22–25,29,31,52]. Information about task duration, which includes the length of the task and the frequency of updates about time remaining, might influence the adaptation of gait features to reduce energy cost, which is considered to be the cost function in the later stages of adaptation [22,27,28,33]. Here, we manipulated the information about task duration during split-belt walking adaptation in neurotypical young adults by comparing two groups: one

with complete certainty about task duration due to accurate information and regular updates of time remaining, and one with high uncertainty about task duration due to inaccurate information and no updates of time remaining. Our results partially supported our hypotheses: while we found differences in adaptation between individuals who knew about a short task duration compared to those who planned to adapt for 30 minutes, the mechanisms underlying these differences contradict our hypotheses. We found that individuals who had detailed, true information about task duration (True group) had a greater increase in metabolic cost from baseline, generated more positive work with the slow leg, and less negative work with the fast leg compared to individuals who did not have detailed information about task duration (Control and False groups). The biomechanical strategies adopted in the True group are the opposite of what has been previously reported to reduce energetic cost during split-belt walking [22,34]. These results support the idea that information about task duration influences the adaptation of walking patterns towards patterns with different energetic costs.

We observed a strong relationship between gait biomechanics and energetic cost in individuals who believed they would sustain the task longer (False group), supporting our second mechanistic hypothesis that individuals may adapt toward a lower cost to maintain an energetic reserve [37]. Surprisingly, the biomechanical strategies that supported this energetic reduction differed from those we hypothesized. We originally expected to see more positive asymmetries due to longer steps with the fast leg, which also allows for more negative work and less positive work by the fast leg [22,34]. We found that the reduction in metabolic cost in the False group correlated with step length asymmetry [22], but this correlation stemmed from changes in the step lengths of the slow leg, not the fast leg. Specifically, the shortening of the slow step length during adaptation was proportional to the reduction in metabolic cost. This is an interesting finding since linear models indicated that, at the group level, there was no change in the slow leg's step lengths from EA to LA. It might be the case that the slow leg adjustments are heterogeneous across individuals, such that there is no group-level effect. Adjustments in step length with the slow leg, rather than the fast leg, may result in a smaller energetic penalty [31], and could supplement other adjustments previously reported [22]. Based on this discussion, we assessed the changes in step lengths; we observed that both the change in slow leg step length ($r = 0.55$, $p = 0.017$) and the change in fast leg step length ($r = −0.68$, $p = 0.002$) were correlated with the change in positive work by the fast leg in the False group. In the True group, only the change in fast leg step length was correlated with the change in positive work by the fast leg ($r = −0.77$, $p < 0.001$). In the Control group, there was no association between the change in step lengths and the change in positive work by the fast leg. These results support the idea that the added slow leg adjustments aim at reducing work and, subsequently, metabolic energy during adaptation.

Our results assessing the exploration of walking patterns were surprising. We observed greater variability in step length asymmetry upon initial split belt exposure in the True and False groups compared to the Control group (Fig 6A). However, this occurred only for step length asymmetry and not for step width or work, indicating that there might be another difference in participant characteristics that potentially influenced these findings. Similar to previous literature [31], we found reductions in variability indicating reductions in exploration across different domains from EA to MA and LA. However, between groups, we found that from MA to LA variability was not different between groups, indicating both similar duration and magnitude of exploration of step length asymmetry over the 10 minutes of adaptation. Thus, our results indicate that information about a longer or shorter task duration did not influence exploration of walking patterns systematically.

Based on our results, a question regarding the True group arises: What cost function was prioritized during split-belt adaptation? Studies have shown that other cost functions can be prioritized over metabolic costs, such as muscle activations [53], comfort [54], time [55], or balance [56]. Since the speed of the belts and the trial duration are fixed, no strategy would reduce task duration as in other studies [55,57,58]. Another potential explanation relates muscle recruitment and power generation being regulated based on expected exercise duration [37,38], with muscle recruitment and power generation directly influencing energetic consumption [59,60]. An example of this anticipatory regulation can be seen when cyclists increase power generation as the distance to the endpoint decreases [61,62]. Furthermore, if the distance to the endpoint is unknown, cyclists will maintain lower power generation than when these same participants knew the distance

to the endpoint [62]. In another study on runners, individuals who believed they would run for longer had lower perceived exertion throughout the task compared to those who were informed of a shorter task duration [62]. While we also did not observe a clear end spurt [39], we saw increased work of the slow leg throughout adaptation, which might indicate that the entire 10-minute task was performed as the end spurt due to the short task duration and moderate effort of the split-belt task. Another alternative explanation is that the cost functions in the True group are the same, mainly balance and energetic cost, but the time during adaptation, where each cost function shapes the gait pattern, differs from what has been previously reported [25,28]. Based on our step width results as a proxy for balance (Fig 2C), while not significant, there appears to be a visually apparent difference in step widths among participants in the True group, with participants in this group showing wider steps than those in the Control and False groups. Thus, it could be the case that for a short time duration and detailed task information, energetic cost becomes less of a priority, and other cost functions are optimized over a longer timescale.

Another key and surprising finding of our study is that information about task duration (which includes duration of task and frequency of updates) influenced the aftereffect of the split-belt walking pattern. Previous work has shown that split-belt aftereffects during post-adaptation can vary in different pathological populations [63–65], when adapting through gradual adjustments of the belt speeds and a longer adaptation time [66], or when using visual feedback [15]. Similarly, less frequent updates have been shown to result in greater retention during other types of motor adaptation tasks [35]. Here, we found that individuals who received false information about a longer task duration experienced a greater aftereffect when the belts returned to moving at the same speed. This aftereffect was measured as a more positive step length asymmetry during EP, due to the step lengths on the right leg remaining shorter in the False group compared to the Control and True groups. These shorter steps on the right leg were consistent with the strategy that supported energetic cost reduction during split-belt walking in the False group. Two potential mechanisms might explain this aftereffect. On one hand, the increased aftereffect in the False group might indicate an initial selection of a previously experienced more optimal gait pattern, supporting the idea that individuals select these patterns even for small energetic savings [32]. On the other hand, it may be that the False group, who not only received false information at the beginning of the task but also received no updates during the split-belt task, adapted in a more implicit manner, as less awareness was directed to the task compared to the True and Control groups; known to increase retention and transfer of the adapted gait pattern [66,67].

Our study has several limitations. First, many possible types of information could have been given to the experimental groups, as we can modify the duration information or frequency of information: we could have had a False group that was told false information about 30 minutes of task duration and given minute by minute updates or we could have had a Control group that walked for 10 minutes with no updates, among many other possible manipulations. We designed our two experimental groups to create the biggest differences in certainty about task duration that would influence behaviors during locomotor adaptation. We designed our Control group to mimic what was done in our previous studies, allowing us to assess potential confounders in our experimental setup that might have resulted in our surprising findings. Future studies can assess whether the differences between groups are due to the duration alone, or due to the frequency of the regular updates. Another limitation is that we did not systematically assess fitness levels across participants. While our criteria excluded people who, at the time of the experiment, exercised more than 10 hours per week, we did not exclude sedentary participants or those with a prior history of participation in high-performance sports. As such, we cannot determine if either group had different levels of fitness, which has been shown to affect energetic expenditure [68] and adaptation [24]. This seems to be the case for the Control group, which exhibited a lower level of metabolic cost during split-belt adaptation. We controlled for these differences in our statistical analyses by using baseline costs as a covariate. Previous research has shown that the amount of mechanical work reduction during adaptation is influenced by the timing of work generation during the pendular phase of gait [19,20]. Our study did not assess whether our experimental manipulation affected this pendular work generation. The different contexts created by the various types of instructions in each group

may have led to differences in autonomic responses. While we measured heart rate, we were unable to obtain measures of heart rate variability to determine sympathetic activation during the task in each group, which could explain differences in heart rate and metabolic cost responses [69]. Finally, we used step width as a proxy for balance since our marker set did not allow for the assessment of more sensitive measures of balance during adaptation, such as extrapolated center of mass [25,46] or whole body angular momentum [26,70], which may have hindered our ability to measure additional balance adjustments in our protocol.

## Conclusion

Humans continuously adapt their locomotor patterns. Adaptation of kinematics, kinetics, and metabolic cost has been reported during split-belt walking protocols of varying durations. However, it is still unknown whether these adapted gait patterns are influenced by different adaptation durations. We manipulated information about task duration during split-belt walking in young adults. We found that when individuals receive detailed information about a short task duration, they perform the task in an effortful and energetically costly manner, and the changes in energetic cost are not associated with changes in gait biomechanics. In contrast, the strength of the association between changes in energetic cost and changes in biomechanics seems to depend on uncertainty regarding task duration. We also found that information about longer task duration led to greater aftereffects during post-adaptation, indicating that planning to sustain a task for longer can influence the retention of walking patterns. The simplest implication of our findings is that adapted locomotor and energetic patterns are influenced by information about task duration; as such, task duration information should be controlled and clarified during experimental paradigms of motor adaptation. Our findings also imply that information about task duration can be manipulated to elicit different efforts and varying retention of adapted patterns, which can have implications for movement interventions in patient populations, particularly individuals with endurance limitations. Future studies should investigate how feedback about task duration influences adaptation, effort generation, and retention of walking patterns in pathological populations.

## Supporting information

**S1 File. Results for heart rate, RPE, and negative work during adaptation as well as metabolic cost and work during post-adaptation are presented in the Supporting Information.**
(DOCX)

## Acknowledgments

We would like to thank Dr. Aram Kim for helpful manuscript reviews. We would like to thank Drs. Lorcan Graham, Makenna Howard, Julia Malloy, Khloe McCarthy, Marissa Vasquez, Laura Corona, Jessica Cota, Emi Heisterkamp, Brandon Nakano, Rebecca Wingen, and Joseph Yeung for helping with data collection for this study. We would also like to thank Alejandro Aguirre-Ramirez and Hailey Hashiguchi for helping with data collection and labeling. Finally, we would like to thank all participants who volunteered for this study.

## Author contributions

**Conceptualization:** Andrew Hooyman, Russell Johnson, Natalia Sanchez.

**Data curation:** Samantha Jeffcoat, Adrian Aragon.

**Formal analysis:** Samantha Jeffcoat, Adrian Aragon, Andrian Kuch, Natalia Sanchez.

**Funding acquisition:** Natalia Sanchez.

**Investigation:** Samantha Jeffcoat, Adrian Aragon, Andrian Kuch, Natalia Sanchez.

**Methodology:** Shawn Farrokhi, Andrew Hooyman, Russell Johnson, Natalia Sanchez.

**Project administration:** Samantha Jeffcoat, Adrian Aragon.

**Software:** Samantha Jeffcoat, Andrian Kuch, Andrew Hooyman, Natalia Sanchez.

**Supervision:** Natalia Sanchez.

**Visualization:** Samantha Jeffcoat, Shawn Farrokhi, Natalia Sanchez.

**Writing – original draft:** Samantha Jeffcoat, Natalia Sanchez.

**Writing – review & editing:** Samantha Jeffcoat, Adrian Aragon, Andrian Kuch, Shawn Farrokhi, Andrew Hooyman, Russell Johnson, Natalia Sanchez.

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
