## [Decision Letter · Decision Letter 0]

22 Sep 2025

Dear Dr. Sanchez,

Thank you for submitting your manuscript to PLOS ONE. After careful consideration, we feel that it has merit but does not fully meet PLOS ONE’s publication criteria as it currently stands. Therefore, we invite you to submit a revised version of the manuscript that addresses the points raised during the review process.

We look forward to receiving your revised manuscript.

Kind regards,

Charlie M. Waugh

Academic Editor

PLOS ONE

Journal Requirements:

[NCMRR R03HD107630, NCATS R03TR004248 to N. Sanchez]. 

3. Please expand the acronyms “NCMRR” and “NCATS” (as indicated in your financial disclosure) so that it states the name of your funders in full.

Reviewers' comments:

Reviewer's Responses to Questions

**Comments to the Author**

1. Is the manuscript technically sound, and do the data support the conclusions?

Reviewer #1: Partly

Reviewer #2: Yes

2. Has the statistical analysis been performed appropriately and rigorously?

Reviewer #1: Yes

Reviewer #2: Yes

3. Have the authors made all data underlying the findings in their manuscript fully available?

Reviewer #1: Yes

Reviewer #2: Yes

4. Is the manuscript presented in an intelligible fashion and written in standard English?

Reviewer #1: Yes

Reviewer #2: Yes

Reviewer #1: This study by Jeffcoat and colleagues investigates the idea that information about task duration – both in expected duration, and also in during-task time updates – could affect gait adaptation. The authors studied three groups of young adults – those who knew the task would be 10 minutes and received an update after 5 minutes (Control); those who knew the task would be 10 minutes and received updates every minute (True); and those who expected the task to take 30 minutes and received no time updates (False). The authors offered competing mechanistic hypotheses as to why the False group would differ from the other two groups; however, the main findings indicate differences between the True group and the other two groups, rather than between the False group and the other two groups. These findings have implications for understanding the role of perceived task time in implicit locomotor learning.

General:

While I think this is an interesting study and the findings certainly hold value to the field, I think that the study design limits its interpretations. The effect of participants’ expected task duration (False versus Control) may be confounded due to the fact that elapsed time updates were not held constant between the Control and False groups, and it appears from the results of this study that the frequency of task duration updates have significant effects on energetic measures. Unless there is evidence that the Control group’s 5-minute time elapsed update would not be a confound when comparing to the False group’s no-update (which, lines 592-595 of the discussion lead me to believe is unlikely), I am hesitant to believe this is a reasonable comparison. In addition, the comparison between the True and False groups is complex and not as binary as the group names suggest, as the elapsed duration updates were also not held constant between these two groups so the resulting differences would be due to some combination of expected duration and the frequency of elapsed duration updates.

Ideally, this would be solved by collecting data on participants that a) expected 10 minutes and walked 10, and received a no updates (new Control, to compare Control/False), and/or b) expected 30 minutes but walked 10, and received minute-by-minute updates (to compare True/False). However, I understand that resources are limited, and this is not always possible. At the very least, I believe a stronger emphasis throughout the paper on how the “True” group is different from the Control group, in addition to background literature on the effects of frequent versus infrequent time updates on learning, is warranted. And, when comparing True and False groups, caution in interpretation is advised.

Introduction:

Line 92-94: Can the authors please provide evidence that information about task duration affects other motor learning tasks? i.e., what is the supporting evidence behind this rationale?

Line 103: Did the authors have any hypotheses about difference between the Control and the True groups – based on differences in the timing of elapsed duration updates?

Methods:

Line 129: I appreciate the a priori sample size calculation.

Line 139-147: I understand why the authors decided to collect a control group for comparison to other works. However, why did the True group receive minute-by-minute updates, and the False group receive none? It seems that the results will not necessarily be due to expected duration, but to some combination of expected duration and the timing of elapsed duration updates. Why not also give minute-by-minute updates to the False group?

Line 144: these references are not in the same format as the rest of the manuscript.

Results:

Line 425: Which hypothesis does this support? Hypothesis 1 was that “the False group will show less adaptation of step length asymmetry, higher metabolic cost, and more work by the legs compared to the True and Control groups,” while hypothesis 2 was that “the False group will show greater adaptation, lower metabolic cost, and less work by the legs compared to the True and Control groups.” As I understand, this finding does not support either of these hypotheses, as the False group did not differ from Control or True group, but instead indicates an important distinction of the True group from the other two groups (along with the metabolic cost findings) – which I don’t believe had been hypothesized.

Figures:

Figure 1: The authors might consider changing the color of either the True group or of the slow belt – the greens look the same.

Figure 2 caption, Line 361: I think this should just say “Abbreviations as in Figure 1.”

Discussion:

Line 565: Isn’t the change in fast-leg step length correlated with the change in positive work by the fast leg in the True group (r=0.77, p<0.001) as well? (From figure 5). Only the relation between change in slow leg step length/change in positive work by the fast leg is unique to the False group. I think this still supports your claim in this sentence but is an important distinction.

Line 577: not only information about a longer versus shorter task, but also frequent task duration information versus infrequent task duration information.

Lines 592-595: This is an interesting explanation of why the True group might have differed from the Control group. However, it also would suggest that the Control group is not an adequate comparison to the False group, since the Control group received information that they were halfway done, while the False group received no updates.

Reviewer #2: I appreciated reading this outstanding manuscript. It was very clear and made for easy reading. The study is carefully designed and rigorously conducted and reported, with a nice statistical sample estimation and nice statistics showing that there aren't much differences between the different participant groups.

Very minor comments below, all of which are typographical/formatting related, and none of which are essential for publication.

The abstract is clear and detailed, but I wonder if it is more detailed than necessary (with N, r and p values and so on). I'm totally fine if the authors want to leave it as is.

Lines 104 and 107. (True group – True task duration information) (False group – False task duration information).

I wonder if you can go (True group, that is, containing true task duration information), etc. May be easier to read than having tha emdash/endash.

Line 115. typo: les  less

Line 235. typo: 'With' should be 'with'

Line 316. extraneous hyphens seem like unconventional typesetting. maybe best to say "N = 14 for the Control group, N = 19 for the True group, and ..."

The first paragraph of the Results section (316-323) seems more like 'methods'. Perhaps move that paragraph entirely to the Methods section?

**Do you want your identity to be public for this peer review?** For information about this choice, including consent withdrawal, please see our Privacy Policy

Reviewer #1: No

Reviewer #2: **Yes: ** Manoj Srinivasan

---

## [Author Response · Author response to Decision Letter 1]

24 Oct 2025

Reviewers' comments:

Reviewer #1: This study by Jeffcoat and colleagues investigates the idea that information about task duration – both in expected duration, and also in during-task time updates – could affect gait adaptation. The authors studied three groups of young adults – those who knew the task would be 10 minutes and received an update after 5 minutes (Control); those who knew the task would be 10 minutes and received updates every minute (True); and those who expected the task to take 30 minutes and received no time updates (False). The authors offered competing mechanistic hypotheses as to why the False group would differ from the other two groups; however, the main findings indicate differences between the True group and the other two groups, rather than between the False group and the other two groups. These findings have implications for understanding the role of perceived task time in implicit locomotor learning.

We would like to thank the reviewer for their assessment of our manuscript and insightful comments. Upon careful review of their comments, a common thread among the reviewer’s comments is their concern about the information given to each experimental group, which the reviewer separates into duration and frequency. We would like to clarify that we have combined both duration and frequency of information under the umbrella term ‘information'. While we agree that each of these two constructs may have separate effects, our goal was not to study these separate effects but to combine the type and frequency of information in a way that would maximize differences between the True and False groups. Our edits include making this distinction between information on duration and frequency of updates, and explicitly mentioning the limitation that we cannot assess the separate effects of each. We hope that the response to comments provides clarity about why we chose the different groups and our experimental manipulation.

General:

While I think this is an interesting study and the findings certainly hold value to the field, I think that the study design limits its interpretations. The effect of participants’ expected task duration (False versus Control) may be confounded due to the fact that elapsed time updates were not held constant between the Control and False groups, and it appears from the results of this study that the frequency of task duration updates have significant effects on energetic measures. Unless there is evidence that the Control group’s 5-minute time elapsed update would not be a confound when comparing to the False group’s no-update (which, lines 592-595 of the discussion lead me to believe is unlikely), I am hesitant to believe this is a reasonable comparison. In addition, the comparison between the True and False groups is complex and not as binary as the group names suggest, as the elapsed duration updates were also not held constant between these two groups so the resulting differences would be due to some combination of expected duration and the frequency of elapsed duration updates.

Ideally, this would be solved by collecting data on participants that a) expected 10 minutes and walked 10, and received a no updates (new Control, to compare Control/False), and/or b) expected 30 minutes but walked 10, and received minute-by-minute updates (to compare True/False). However, I understand that resources are limited, and this is not always possible. At the very least, I believe a stronger emphasis throughout the paper on how the “True” group is different from the Control group, in addition to background literature on the effects of frequent versus infrequent time updates on learning, is warranted. And, when comparing True and False groups, caution in interpretation is advised.

We thank the reviewer for the careful assessment of the manuscript. The reviewer brings up multiple points, and we would like to structure our answer to address each point:

1. “The effect of participants’ expected task duration (False versus Control) may be confounded due to the fact that elapsed time updates were not held constant between the Control and False groups, and it appears from the results of this study that the frequency of task duration updates have significant effects on energetic measures”. “Ideally, this would be solved by collecting data on participants that a) expected 10 minutes and walked 10, and received a no updates (new Control, to compare Control/False), and/or b) expected 30 minutes but walked 10, and received minute-by-minute updates (to compare True/False).”

First, we agree with the reviewer comments; many potential control groups could be used in our study. When deciding the protocol for the control group, we discussed the idea of a group that received accurate information of the task duration and no updates during the task; we did not proceed with this group as our previous protocols, and based on anecdotal discussions of the protocols followed by colleagues, updates were usually provided either at the halfway point or as requested by participants. We compared our results with those of the Sánchez 2019 free adaptation task, the Sánchez 2021 short adaptation comparison group, and Brinkerhoff 2024. We confirmed that our results in the control group were consistent with our prior studies.

Unfortunately, as the reviewer noted, we are unable to recruit additional participants, as all study personnel have moved on to subsequent positions. Even with this limitation, we believe the data in N=52 participants still provides a rigorous examination of the effects of task information on motor adaptation in split-belt treadmill walking. However, the point raised by the reviewer is very important and explicitly included in many portions of the manuscript, as shown in the manuscript edits listed below (after item 4).

2. “In addition, the comparison between the True and False groups is complex and not as binary as the group names suggest, as the elapsed duration updates were also not held constant between these two groups so the resulting differences would be due to some combination of expected duration and the frequency of elapsed duration updates”

We designed our groups in a way that maximized the uncertainty about task duration between the groups. This resulted in two potential factors that influenced the differences between the groups: the time information about task duration and the frequency of the information. Both of these factors are summarized in “Information of task duration,” as stated in the title of the manuscript. We chose to study the combination of these two aspects of the information in this manuscript to maximize the differences between groups. Future work can study each of these constructs in isolation. To address the reviewer’s comments, this is now explicitly presented in the introduction.

3. At the very least, I believe a stronger emphasis throughout the paper on how the “True” group is different from the Control group,

We have updated our group definitions, in agreement with the reviewer’s comments, to be: True group: True task duration information with frequent updates on time remaining, and False group: False task duration information with no updates of time remaining.

4. Background literature on the effects of frequent versus infrequent time updates on learning, is warranted.

The effect of time updates on motor adaptation for split-belt walking has not been specifically studied; therefore, we believe this manuscript will contribute to the existing literature. We present information regarding feedback schedules and their influence of adaptation, as well as time information and their influence on effort. These combined knowledge is what shaped our hypotheses. We reiterate that Given that the influence of task duration information on split-belt adaptation has not been directly studied, our work offers a novel contribution to the motor adaptation literature by exploring how time information may shape locomotor adaptation. We include this information in the introduction as show below.

To address points 1-4, we have made the updates listed below throughout the manuscript and explicitly included the reviewer's comments in the limitations section.

Abstract:

Studies of locomotor adaptation have shown that adaptation can occur in short bouts and can continue for long bouts or across days. Information about task duration might influence the adaptation of gait features, given that task duration influences the time available to explore and adapt the aspects of gait that reduce energy cost. We hypothesized that information about task duration and frequency of updates influences adaptation to split-belt walking based on two competing mechanisms: individuals anticipating a prolonged adaptation period may either (1) extend exploration of energetically suboptimal gait patterns, or (2) adapt toward a more energy-efficient pattern earlier to maintain an energetic reserve.

Introduction:

Since energetic cost is optimized during the later stages of adaptation, information about a longer task duration might influence the adaptation of the aspects of gait underlying this energetic cost reduction. As individuals adapt their walking pattern to the split-belt treadmill, they reduce step length asymmetry, positive mechanical power, and positive mechanical work generated by their legs [11,19,20,22,24,25], which is associated with a reduction in metabolic cost [11,22]. In fact, studies have shown that walking patterns can change continuously, over different timescales, even for small energetic savings [32–34]. Thus, it may be the case that planning to sustain a task for a shorter or longer time may influence how the walking pattern is adapted to reduce energy cost.

In addition to task duration itself, the amount and frequency of information provided about that duration may also influence motor adaptation and control. For example, studies on feedback schedules have shown that infrequent feedback can impair immediate performance but enhance long-term learning [35]. Although our study does not provide performance feedback, it is possible that infrequent updates about task duration could produce similar effects. Moreover, motor control studies have demonstrated that when participants are given information about time constraints, they often select more effortful, energetically sub-optimal walking speeds to ensure task completion [36]. While in our study the time and speeds are fixed, information about time duration may still influence effort. This idea aligns with evidence that the central nervous system modulates muscle recruitment and power output based on expected exercise duration [37,38], and with the "end spurt" phenomenon, where effort increases as individuals approach the end of a task [39,40]. Given that the influence of task duration information on split-belt adaptation has not been directly studied, our work offers a novel contribution to the motor adaptation literature by exploring how time information and frequency of time updates may shape locomotor adaptation.

The primary aim of our study was to determine if information about task duration affects the adaptation of locomotor patterns. We tested the hypothesis that individuals provided with minute-by-minute information of a 10-minute adaptation duration (True group – True task duration information with frequent updates on time remaining) will have a different adapted locomotor pattern with different metabolic cost than those individuals who were informed that they will sustain a locomotor adaptation task for a prolonged time of 30 minutes but who will actually sustain it for 10 minutes (False group – False task duration information with no updates of time remaining). We compared our two experimental groups to a Control group, who received true information about task duration before the start of the adaptation trial, without the minute-by-minute updates and with only a time update at the halfway point, consistent with previous work [22,24–26,34]. Two contrasting mechanisms can explain this metabolic and locomotor pattern differential: Mechanism 1) given that the split-belt task is only of moderate intensity, the False group might spend more time exploring suboptimal locomotor patterns as they prepare to sustain the task for longer, whereas the True group will aim to reach a less costly pattern within the allotted time; if mechanism 1 drives adaptation, we hypothesize that the False group will show less adaptation of step length asymmetry, higher metabolic cost, and more work by the legs compared to the True and Control groups. Under mechanism 1, we also hypothesize that the True group will be less costly than the Control group. Mechanism 2) The False group might adapt toward a generally less costly pattern than the True group to be able to maintain an energetic reserve needed to sustain the task longer [36,37,40]; if mechanism 2 drives adaptation, we hypothesize that the False group will show greater adaptation, lower metabolic cost, and less work by the legs compared to the True and Control groups. Under mechanism 2, we expect no differences between the True and Control groups. Since our previous work showed that adaptation duration influences the duration but not the magnitude of the aftereffects during post-adaptation [22], and all groups will adapt for the same duration, we hypothesize that the locomotor pattern during post-adaptation will not differ between groups. A better understanding of how information about task duration and updates provided during adaptation tasks influences adaptive processes is vital in the design of training schedules aimed at retraining walking behaviors as part of rehabilitation interventions [41,42] or via the use of assistive devices [29,33,43,44].

Materials and Methods:

To assess differences between the True group and the False group relative to the locomotor pattern described in prior adaptation studies, we also collected a Control group post-hoc, which, like previous work, was informed about a 10-minute task duration at the beginning of the split-belt adaptation trial and only received one update halfway during adaptation. We collected a Control group of N=14 participants, with a sample size similar to previous studies that showed a reduction in energetic cost or mechanical work during adaptation (N=14 in [45], N=11 in [11] and N=14 in [19]). This group also allowed us to ensure that any potential differences in our True and False groups are due to our experimental manipulation of providing different information on task duration and frequency of updates in the False or True groups, and not due to potential confounding effects of our experimental environment or analyses.

The reference level for the linear models for group was set as Control group, as we are comparing the True and False groups to a protocol that is comparable to that used in the split-belt literature. The reference level for the linear models for time was set as EA for the split-belt models and EP for the post-adaptation models.

Discussion:

Studies of motor adaptation have shown that adaptation begins within the first minute and continues even after 45 minutes of adaptation to split-belt walking [22–25,29,31,51]. Information about task duration, which includes the length of the task and the updates about time remaining, might influence the adaptation of gait features to reduce energy cost, which is considered to be the cost function in the later stages of adaptation [22,27,28,33]. Here, we manipulated the information about task duration during split-belt walking adaptation in neurotypical young adults by comparing two groups: one with complete certainty about task duration due to accurate information and regular updates of time remaining, and one with high uncertainty about task duration due to inaccurate information and no updates of time remaining. Our results partially supported our hypotheses: while we found differences in adaptation between individuals who knew about a short task duration compared to those who planned to adapt for 30 minutes, the mechanisms underlying these differences contradict our hypotheses. We found that individuals who had detailed, true information about task duration (True

---

## [Decision Letter · Decision Letter 1]

19 Nov 2025

Information about task duration influences energetic cost during split-belt adaptation and retention of walking patterns post-adaptation

PONE-D-25-27310R1

Dear Dr. Sanchez,

We’re pleased to inform you that your manuscript has been judged scientifically suitable for publication and will be formally accepted for publication once it meets all outstanding technical requirements.

Kind regards,

Charlie M. Waugh

Academic Editor

PLOS ONE

Additional Editor Comments (optional):

Reviewers' comments:

Reviewer's Responses to Questions

**Comments to the Author**

Reviewer #2: All comments have been addressed

2. Is the manuscript technically sound, and do the data support the conclusions?

Reviewer #2: Yes

3. Has the statistical analysis been performed appropriately and rigorously?

Reviewer #2: Yes

4. Have the authors made all data underlying the findings in their manuscript fully available?

Reviewer #2: Yes

5. Is the manuscript presented in an intelligible fashion and written in standard English?

Reviewer #2: Yes

Reviewer #2: Thank you for addressing all the reviewer comments thoughtfully, both my minor ones and the other reviewer's more detailed feedback. No other comments from me.

**Do you want your identity to be public for this peer review?** For information about this choice, including consent withdrawal, please see our Privacy Policy

Reviewer #2: **Yes: ** Manoj Srinivasan

---

## [Editor Report · Acceptance letter]

PONE-D-25-27310R1

PLOS ONE

Dear Dr. Sanchez,

I'm pleased to inform you that your manuscript has been deemed suitable for publication in PLOS ONE. Congratulations! Your manuscript is now being handed over to our production team.

Kind regards,

on behalf of

Dr. Charlie M. Waugh

Academic Editor

PLOS ONE